# Relationship among Body Composition, Adipocytokines, and Irisin on Exercise Capacity and Quality of Life in COPD: A Pilot Study

**DOI:** 10.3390/biom13010048

**Published:** 2022-12-27

**Authors:** Giuseppina Cuttitta, Maria Ferraro, Fabio Cibella, Pietro Alfano, Salvatore Bucchieri, Angelo Maria Patti, Rosalba Muratori, Elisabetta Pace, Andreina Bruno

**Affiliations:** 1Institute of Translational Pharmacology (IFT), National Research Council (CNR), 90146 Palermo, Italy; 2Institute for Biomedical Research and Innovation (IRIB), National Research Council (CNR), 90146 Palermo, Italy; 3Department of Health Promotion Sciences Maternal and Infantile Care, Internal Medicine and Medical Specialties (PROMISE), University of Palermo, 90127 Palermo, Italy; 4Azienda Sanitaria Provinciale di Palermo, Via Giacomo Cusmano, 24, 90141 Palermo, Italy

**Keywords:** adipocytokines, adiponectin, body composition, COPD, haptoglobin, irisin, leptin, quality of life, 6 min walking test

## Abstract

Adipose tissue is an endocrine organ that interferes with the severity of chronic obstructive pulmonary disease (COPD). Although inflammatory markers, body composition, and nutritional status have a significant impact on pulmonary function, the real contribution of adipocytokines and myokines in COPD is still controversial. We aimed to evaluate the role played by the body composition, leptin, adiponectin, haptoglobin, and irisin on the functional exercise capacity, respiratory function, and quality of life (QoL) in COPD. In 25 COPD (20% GOLD-1; 60% GOLD-2; 20% GOLD-3) patients and 26 matched control subjects, we find that leptin, total adiponectin and haptoglobin are significantly increased whereas the 6 min walk test (6MWT) and physical functioning scores are significantly decreased in COPD versus controls. A significant positive relationship is found between leptin and fat mass and between 6MWT and the good health indicators of nutritional status. A significant inverse relationship is found between 6MWT and leptin and fat mass, FEV_1_ and haptoglobin, and irisin and haptoglobin. Phase angle and leptin level are significant predictors for functional exercise capacity assessed with 6MWT. Taken altogether, the results of this pilot study further support the role played by body composition and adipocytokines on exercise capacity respiratory function and QoL in COPD.

## 1. Introduction

Adipose tissue was historically considered as a passive tissue for energy storage even though it is now widely recognized as an endocrine organ producing a wide variety of bioactive molecules, most of them with a systemic inflammatory effect, named adipocytokines [1]. The latter may act via autocrine, paracrine or endocrine signaling, by interfering also with the pulmonary function and contributing to the pathogenesis of the development of chronic lung diseases [2]. 

Chronic obstructive pulmonary disease (COPD) is the third leading cause of death in the world [3] and is characterized by progressive persistent airflow limitation that is not fully reversible. It belongs to the main four Non-Communicable Diseases (NCDs), a group of chronic inflammatory diseases responsible for premature mortality worldwide, including cardiovascular diseases, cancer, metabolic diseases, and pulmonary disease.

NCDs are included in the Sustainable Development Goals (SDGs) as targets that aim to reduce premature mortality from NCDs by a third by 2030, relative to 2015 levels, and to improve mental health and quality of life (QoL) of these patients [4]. Fatigue is the subjective feeling of tiredness or exhaustion and is one of the most common and distressing symptoms experienced by patients with COPD [5]. Furthermore, symptoms of depression and/or anxiety merit specific enquiry when obtaining the medical history because they are common in COPD [6] and are associated with a poorer health status, increased risk of exacerbations, and emergency hospital admission [7]. Another feature of COPD is the presence of comorbidities contributing to systemic effects of the disease [8,9] that can be amplified by metabolic disorders and altered nutritional status. 

Body composition and nutritional status are well known to be key factors involved in the management of COPD patients as they are associated with a higher disease severity, mortality, and exacerbation risk incidence [10]. In this regard, it is recognized that obesity is an important factor in patients with COPD, due to the low-grade state of systemic inflammation determined by the production of the adipocytokines as leptin, adiponectin, and haptoglobin [11,12,13].

Leptin is one of the most relevant pro-inflammatory adipocytokines since it increases TH1 cytokines and suppresses TH2 cytokine production. It is a pleiotropic hormone that plays a key role in immunometabolism and in inflammation, as its receptors are widely expressed in cells of the innate and adaptive immune systems [14,15,16] as well as in human bronchial, alveolar macrophages, and alveolar epithelial cells, bronchial smooth muscle cells, and bronchial submucosa [17,18,19,20]. Leptin itself is expressed by human lungs, including bronchial epithelial cells and alveolar type II pneumocytes and macrophages [21,22]. These findings suggest that the lung is a target organ for leptin signaling and deregulated signaling can be associated to lung diseases [23]. Our previous study assesses that an increased leptin expression in bronchial mucosa of COPD patients is associated with airway inflammation and airflow obstruction [19]. Another study concludes that leptin circulating levels, corrected for fat mass, is significantly elevated in COPD patients during acute exacerbation [24]. 

Adiponectin is another adipocytokine with a potential role in regulating the inflammatory response in COPD, since it induces the production/release of IL-8 by airway epithelium [25]. It exists as a multimeric complexes with different physiological activities in metabolism and immune systems, including high molecular weight complexes (HMW adiponectin) involved in airway pathophysiology: levels of adiponectin are found more elevated in patients with COPD than in controls and in exacerbated COPD than in stable COPD [26] and serum adiponectin concentrations are found positively related to deaths from respiratory causes [27]. 

Haptoglobin, the main carrier of free hemoglobin involved in the hepatic acute phase response, exerts immunomodulatory functions and represents a typical marker of inflammation by promoting the expression of specific dendritic cells maturation markers and pro-inflammatory Th1-associated cytokine expression [28,29]. Haptoglobin perfectly sits in the intersection between obesity and inflammation as it is also produced by white adipocytes. The lungs are a major site of the extrahepatic synthesis of haptoglobin, and serum haptoglobin levels are found increased in COPD patients [30]. 

As part of body composition, together with obesity, skeletal muscle dysfunction and impaired muscle mass and strength also play an important role for extrapulmonary comorbidity in COPD [31]. Altered body composition is an independent factor in daily life and in important outcomes, such as exercise capacity and inflammation management, in COPD patients. A recent study [32] enrolled 270 stable COPD patients with normal body composition, with obesity, with sarcopenia, and with sarcopenic-obesity. These subjects were tested for exercise capacity, for respiratory muscle strength and for dyspnea severity and for symptoms of anxiety and depression. The authors found that the subjects with sarcopenic-obesity were the most impaired for physical activity assessed by the 6 min walking test (6MWT) and together with the subjects with sarcopenia with the worst muscle strength. Interestingly, the authors found no differences regarding symptoms of anxiety and depression.

In this context, the muscle-derived cytokines, named myokines, as irisin, are found dysregulated in the presence of the cigarette smoking factor, that, in turn, could impair muscle protein synthesis in COPD [33,34].

Lastly, gastrointestinal dysfunction and its symptoms are prevalent extrapulmonary systemic manifestations of COPD associated with reduced QoL and also increased with increasing disease severity [35] as well as gut microbiota which represents a key factor in immune response regulation also in COPD [36]. In this context, the intestinal fatty acid binding protein (I-FABP) is a marker of intestinal integrity [37]. It is a 15-kDa, cytosolic, water-soluble protein, mainly present in the mature enterocytes of the small intestine that quickly diffuses once the integrity enterocyte membrane is lost. The impact of the adipocytokine and of dysbiosis marker dysregulated expression/levels together with the nutritional status parameters in the COPD assessment are still unclear.

To our best knowledge, no previous studies have been conducted on the role played by body composition and the aforementioned cytokines and I-FABP on the functional exercise capacity, respiratory function, and QoL of COPD patients, therefore, we conducted this pilot study to better identify their role in the pulmonary function and in mental health and quality of life of COPD patients in comparison to the matched healthy subjects. 

## 2. Materials and Methods

### 2.1. Study Design and Subjects Enrolled

All subjects were recruited (mean age 67.6 years ± 8.4 SD; 22 F, 29 M) in a period of 12 months and included 25 patients with COPD (20% GOLD STAGE 1; 60% GOLD STAGE 2; 20% GOLD STAGE 3) and 26 healthy matched subjects (controls). COPD patients were derived from outpatient pulmonary service at the National Research Council (CNR), a secondary care referral center in Palermo, Italy. The diagnosis of COPD was based on clinical history and a post-bronchodilator forced expiratory volume in the first second (FEV_1_)/forced vital capacity (FVC) ratio lower than 0.70. The classification of COPD severity followed the GOLD (Global initiative for Chronic Obstructive Lung Disease) guidelines (2022 Report; available from: https://goldcopd.org/wp-content/uploads/2021/12/GOLD-REPORT-2022-v1.1-22Nov2021_WMV.pdf) (accessed on 15 November 2022). All patients with COPD were in a stable state at study entry: no exacerbation history in the last three months that required a change in medication or hospital admission. None of the patients were undergoing long-term oxygen therapy. Healthy control subjects were matched by age and BMI, keeping the sex distribution not significantly different between COPD and control subjects. Controls exhibited an FEV_1_/FVC ratio ≥ 0.70 and a FEV_1_ ≥ 80% of the predicted value. Subjects with a known malignant tumor, with orthopedic, neurologic or unstable cardiac diseases, active inflammatory disease (such as asthma, bronchiectasis or other non-COPD-related disease), decreased comprehension due to conditions such as dementia, or severe mental or physical disabilities that might interfere with the results were excluded. Smoking status included: never smokers (<100 cigarettes in a lifetime), former smokers (quit > 12 months before data collection), and current smokers (quit < 12 months before enrollment) or currently still smoking ≥ 10 cigarettes/day for the past year). Smoking pack years were defined by multiplying the number of packs of cigarettes (1 pack = 20 cigarettes) smoked per day by the number of years smoked; years of smoking were directly garnered during the assessment. For the aims of the present work, current and former smokers were pooled together. Data were collected from structural interviews, clinical examinations, laboratory investigations, and measurements of physical parameters. The procedures of the study included the following: (1) patient’s medical record: age, race/ethnicity, average number of COPD exacerbations per year (based on data within the last two years), smoking history (current or past and pack-years), past medical history, current medications, comorbidities, and hospitalization evaluations; (2) pulmonary function by spirometry test; 3) venous blood samples collection; (4) body composition assessment by bioelectrical impedance analysis (BIA); (5) 6 min walking test (6MWT) (6); administration of the following questionnaires: the 36-item Short Form Survey (SF-36), the Modified British Medical Research Council questionnaire (mMRC), the COPD Assessment Test (CAT), the Beck’s Depression Inventory–II (BDI–II), and the State-Trait Anxiety Inventory-Y (STAI-Y) questionnaires which were evaluated from GPs’ files. A patient card containing any history of hospitalizations and possible comorbidities was provided by a general medical practitioner to the interviewer for each patient that provided consent for all the procedures. The chronic medical conditions for all patients enrolled in the study were compiled and counted, including those grouped into cardiovascular or respiratory diseases. The study fulfilled the criteria of the reference Ethics Committee of Policlinico-Giaccone Hospital, Palermo, Italy (No. 10/2013, 18 September 2013) and was in agreement with the Helsinki Declaration. All subjects gave their written informed consent. The individual privacy of clinical data was guaranteed under Italian law. The complete sequence of data collection from the subjects who participated in the study is depicted in the flow chart of Figure 1.

### 2.2. Bioelectrical Impedance Analysis (BIA)

To determinate body composition, all subjects recruited underwent BIA by using a single frequency (50 kHz) analyzer Akern BIA 101 (Akern RJL Systems, Florence, Italy) as previously described [38]. Briefly, subjects were positioned supine on a nonconductive surface, arms and legs abducted at 30° throughout, and electrodes were placed on the right side of the body on the wrist and on the foot [39,40]. Bioelectric impedance (Ohms) is the square root of the sum of the squares of resistance (R, Ohm) and reactance (Xc, Ohm). Based on this concept, BIA allows the determination of direct measurements, as phase angle (PhA, degrees), body cellular mass (BCM, Kg) and total body water (TBW, liters), and of indirect and estimated measurements, such as fat mass (FM, Kg) and fat-free mass (FFM, Kg), obtained by the electrical measurements together with anthropometric and the personal subject’s data. TBW included intracellular and extracellular water (ICW and ECW, liters), body mass index (BMI, kg/m^2^), body cellular mass index (BCMI/m^2^), muscle mass (MM, kg), and fat-free mass index (FFMI/m^2^) are also estimated parameters useful to determinate the whole body composition. The most important parameter that represents a direct indicator of healthy composition is the PhA, geometrically expressed as a ratio of R over Xc, that gives direct information on cellular mass amount together with the BCM, which is related to metabolism, muscle, and nervous cells. BIA is a safe, easy, and non-invasive method that gives an individual impedance vector compared with the 50, 75, and 95% tolerance ellipses calculated in the healthy population as reference, allowing evaluation both in physiological and in clinical conditions, including in lung disease patients [41].

### 2.3. Respiratory Functions and Walking Test

Pulmonary function spirometric measurements were performed before and 15 min after inhalation of 400 μg salbutamol by using the spirometry test. Height (cm) and weight (kg) were obtained for all subjects standing in a position without shoes using a stadiometer and an electronic digital scale. Pulmonary function tests were performed with a spirometer (Baires System; Biomedin, Padua, Italy). FEV_1_, FVC, and maximum mid-expiratory flow (FEF_25–75%_) were measured according to ATS/ERS (American Thoracic Society/European Respiratory Society) guidelines [42]: the best FVC and FEV_1_ were retained and FEF_25–75%_ was selected from the maneuver with the largest sum of FEV_1_ and FVC. For each subject, FEV_1_/FVC ratio was also computed. Spirometry predicted values were those from Quanjer et al. [43]. All subjects recruited also performed 6MWT: it is a simple and standardized tool commonly used to assess the functional exercise capacity of patients with chronic respiratory diseases [44]; 6MWT includes the total distance walked by the patient, as well as other relevant cardiopulmonary parameters such as the heart rate (HR), and the oxygen saturation level (SpO_2_ measured before and after the test). It was carried out indoors: a 30 m long corridor with a straight, flat surface. The space was marked by a distance marker every 3 m. At the beginning and after the completion of the 6MWT, each subject was presented with the Borg dyspnea scale [45]. On a scale ranging from “0  =  none” to “10  =  very, very severe,” the subjects assessed their current degree of shortness of breath.

### 2.4. SF-36, mMRC, CAT, BDI–II, and the STAI-Y Questionnaires

QoL of the subjects enrolled in this experimental design was assessed by using an SF-36 questionnaire [46]. There are two distinct concepts measured by the SF-36: a physical dimension, represented by the Physical Component Summary, and a mental dimension, represented by the Mental Component Summary. All scales do contribute in different proportions to the scoring of both Physical and Mental Components measures. The questionnaire, consisting of 36 items, was summarized, and transformed to give eight summary scales measuring health concepts. The Physical Health measures were physical functioning (10 items), role-physical limitation (4 items), bodily pain (2 items), and general health perception (5 items). The Mental Health measures were vitality (4 items), social functioning (2 items), role limitation attributable to both emotional problems (3 items), and mental health (5 items). Dyspnea perception was evaluated by the mMRC [47], a short questionnaire that allows a numeric value to be placed on each subject’s exercise capacity. It uses a scale from 0 to 4: 0, no breathlessness except on strenuous exercise; (1) shortness of breath when hurrying on the level or walking up a slight hill; (2) walks slower than people of the same age on the level because of breathlessness or has to stop to catch breath when walking at their own pace on the level; (3) stops for breath after walking ~100m or after a few minutes on the level; (4) too breathless to leave the house, or breathless when dressing or undressing. It was administered by an interviewer with the statements framed as questions. Severity of COPD symptoms and health status were assessed using CAT Questionnaire [48], containing eight items assessing cough, phlegm, chest tightness, and breathlessness going uphill/stairs, activity limitations at home, confidence leaving home, sleep, and energy. Each item is scored from 0 to 5 with a total score of 0–40, where 20 is the threshold for clinically significant symptoms. Higher scores correspond to poorer health status in COPD patients. Breathlessness is a complex subjective sensation that is an important feature of respiratory disease and depression and/or anxiety symptoms which may occur in COPD patients. In this study, the Italian version of the Beck Depression Inventory (BDI)–II was used [49]. This version consists of 21 items, in which four response options are presented on a scale of 0 to 3. It showed a good internal consistency (Cronbach's alpha 0.89) and good convergent and divergent validity. Its construct validity was also successfully tested by comparing scores with other measures for depression. Lastly, the STAI-Y consists of twenty statements that evaluate how the respondent feels ‘generally’. All items are scored on a 4-point Likert scale (to mean ‘not at all’, ‘somewhat’, ‘moderately’, or ‘very much so’ for the STAI-Y-1; to mean ‘almost never’, ‘sometimes’, ‘often’, ‘almost always’ for STAI-Y-2); scoring is reversed for 10 STAI-Y-1 items and 9 STAI-Y items. The total score for both scales ranges from 20 to 80 (higher scores indicate more severe anxiety).

### 2.5. Adipocytokines, Irisin, and Intestinal Fatty Acid Binding Protein Measurement

In serum, we assessed these following circulating biomarkers by using the enzyme-linked immunosorbent assays (ELISA) method: leptin, total adiponectin, high molecular weight (HMW) adiponectin, and intestinal fatty acid binding protein (I-FABP) (kit from R&D Systems, Minneapolis, MN, USA) together with the assessment of haptoglobin (kit from Cloud-Clone Corp., Buckingham, MK18 1TF, UK) and of irisin (kit from Cusabio, Houston, TX, 77054, USA). Serum samples were collected in fasting condition and stored at −80 °C until assayed. At the end of protocol, the optical density of each well was determined immediately, using a microplate reader (Microplate reader Wallac Victor2 1420 Multilabel Counter, Perkin Elmer, Waltham, MA, USA) [50].

### 2.6. Statistical Analysis

Descriptive statistics were used to characterize sample characteristics using means and standard deviations (SD) or medians and interquartile range (IQR) to describe normally and not normally distributed quantitative variables, respectively. Numbers and percentages were used for categorical variables. One-way analysis of variance (ANOVA) or Mann–Whitney U-test—as applicable—were used to test possible differences in the quantitative variables between groups. Two-way ANOVA was used to explore the influence of sex and groups and their interaction on variables. χ^2^ test was used for testing differences in the distribution of categorical variables. Correlation between variables was evaluated by means of linear correlation, after normalization, in the case of not normally distributed variables. Normalization of not normally distributed variables was performed by mean and standard deviation, thus producing a new variable with normal data distribution presenting the same mean and standard deviation of the original variable. Aimed at correcting for confounder/effect modifier variables, a multiple regression model was built to evaluate the effect of independent variables on 6MWT. A *p* level < 0.05 was considered significant in all the analyses. The statistical analyses were conducted by means of SPSS (IBM SPSS Statistics for Windows, Version 20.0. Armonk, NY, U.S.A., IBM Corp. IBM Corp.).

## 3. Results

### 3.1. Anthropometric and Functional Data

The descriptive anthropometric and functional characteristics of enrolled subjects are presented in Table 1, separately for COPD and controls. Despite BMI being not significantly different between the two groups (COPD and controls), waist and abdomen circumferences and fat mass (Table 2) were significantly higher among COPD (*p* = 0.007, *p* = 0.017, and *p* = 0.046, respectively) (Figure 2A–C), whereas only waist circumference was also significantly higher in males with respect to females (*p* = 0.004). Among the other bioimpedance parameters, no one else was significantly different between the two studied groups (Table 1). As expected, all the pulmonary function variables, expressed as percent of predicted values, were significantly higher in controls than in COPD (*p* = 0.0002 for FVC% and *p* < 0.0001 for FEV_1_%, FEV_1_/FVC%, and FEF_25–75%_%), as well as the distance covered during the 6MWT being shorter among COPD patients (*p* = 0.002) (Table 1). The smoking habit was significantly different between the two groups: among COPD subjects, only 4/25 were never smokers, whereas 14/26 were never smokers among controls (*p* = 0.02). Figure 2 also shows the results of the linear regression analysis between waist circumference, abdomen circumference, and fat mass to the adipocytokine leptin, normalized by mean and SD (Figure 2D–F). All the correlations were highly significant (*p* = 0.0002, *p* < 0.0001, and *p* < 0.0001, respectively).

Adipocytokines, irisin, and I-FABP circulating levels. Two-way analysis of variance for the effects of group and sex: for all the variables, with the exception of haptoglobin (values as mean ± standard deviation), two-way ANOVA (values as median and interquartile range) was performed after data normalization by mean and standard deviation. HMW = high molecular weight; I-FABP = intestinal fatty acid binding protein.

### 3.2. Data from Questionnaires 

In Table 1, we also reported the results relevant to questionnaires. SF-36 questionnaires for the quality of life provided not significant differences for both Physical Health section (median (IQR)) 48.0 (41.5–53.0) in COPD and 50.0 (48.0–53.0) in controls (*p* = 0.19, Mann–Whitney U-test), and Mental Health section 47.0 (36.5–51.5) in COPD and 45.5 (33.0–50.0) in controls (*p* = 0.42), whereas, interestingly, the SF-36 Physical Functioning section provided a significant lower score in COPD than in controls (*p* = 0.004). As expected, dyspnea perception mMRC questionnaire gave a significantly higher score in COPD than in controls (*p* < 0.0001). The other questionnaires evaluated were not significantly different between the two groups.

### 3.3. Adipocytokines, Irisin, and I-FABP Evaluations

Leptin, leptin normalized for fat mass (leptin/fat mass ratio), and total adiponectin were significantly higher in COPD than in controls, both for categories (*p* = 0.003, *p* = 0.001, and *p* = 0.04, respectively) and for sex effects (significantly higher in females than in males, *p* = 0.007, *p* < 0.0001, and *p* = 0.02, respectively) (Table 2, Figure 3A–C). Haptoglobin was significantly higher among COPD subjects than in controls (*p* = 0.006), only for group effect but not for sex effect (Figure 3D). Irisin, irisin normalized for muscle mass (irisin/muscle mass ratio), and I-FABP were not significantly different between the two evaluated groups. Lastly, irisin/muscle mass ratio was significantly higher among non-smokers than in smokers (*p* = 0.045), and in females than in males (*p* = 0.0009) (Figure 3E).

### 3.4. Correlation Data among the Evaluated Variables

6MWT was significantly positively correlated to the good health BIA parameters PhA, BCM, MM, and ICW (Figure 4A–D) whereas it was significantly and negatively correlated to fat mass, leptin, leptin/fat mass ratio, and HMW adiponectin (Figure 4E–H). The pulmonary function variable FEV_1_ (as percent of predicted value) presented a strong significant negative linear correlation with haptoglobin (*p* = 0.0005) (Figure 5A). Furthermore, only in the COPD group, but not in controls, haptoglobin normalized for fat mass (haptoglobin/fat mass ratio) was significantly and negatively correlated to irisin and to irisin/muscle mass ratio (*p* = 0.002 and *p* = 0.003, respectively) (Figure 5B,C), whereas total adiponectin was significantly and negatively correlated to irisin, irisin/muscle mass ratio, and to the Physical Functioning score of SF-36 questionnaire (*p* = 0.012, *p* = 0.02, and *p* = 0.025, respectively) (Figure 5D–F). Lastly, in a multiple linear regression model (Table 3) for 6MWT (R^2^ = 0.701), when corrected for sex, age, and FEV_1_ as percent of predicted, leptin, and phase angle levels were strongly significant predictors (*p* <0.0001 for both). Interestingly, in the same model, the variable “group” (i.e., COPD or controls) was not significant. 

## 4. Discussion

Adipose tissue has been implicated in the regulation of the immune system, also in inflammatory chronic diseases such as COPD [1,2,20,23]. Skeletal muscle is also involved in metabolism regulation by myokines’ production [33,34]. Earlier studies suggested that body composition and physical activity represent crucial factors for the respiratory function and quality of life of COPD patients [10].

Furthermore, gut microbiota also represents a key factor in immune response and immune system regulation in COPD, likewise gastrointestinal dysfunction and its symptoms are prevalent extrapulmonary systemic manifestations of COPD associated with reduced quality of life of these patients [36]. However, whether all these aforementioned factors are functionally involved in the progression and management of COPD disease is a matter of debate. In this pilot study with 51 subjects enrolled, we investigated in COPD patients in comparison to matched healthy subjects, the role of leptin, adiponectin, haptoglobin, irisin, and I-FABP in the body composition assessment, in functional exercise capacity, in respiratory function, and in quality of life. It is important to highlight that, in our experimental design, we used the handheld bioelectrical-impedance analysis (BIA) for body composition evaluation, a valid and accurate tool comparable to the dual X-ray absorptiometry (DXA) in COPD patients, as a recent study identified BIA with a high concordance with DXA for fat mass [51].

The following new main findings are reported:

(1) Adiposity markers such as fat mass, abdomen, and waist circumferences are significantly higher in COPD than in controls;

(2) With the exception of the high molecular weight complexes (HMW) of adiponectin, each of the adipocytokines evaluated (leptin, leptin/fat mass ratio, adiponectin, and haptoglobin) is significantly higher in COPD than in controls, whereas I-FABP is unchanged;

(3) Irisin/muscle mass ratio is significantly higher in non-smokers than in smokers, regardless of belonging to COPD or control groups;

(4) The SF-36 Physical Functioning section questionnaire for the quality of life provides a significant lower score in COPD than in controls;

(5) Leptin is strongly positively correlated with adiposity markers, 6MWT (for the functional exercise capacity of COPD patients) is strongly positively correlated with good health nutritional parameters and strongly inversely correlated with adiposity markers; likewise, FEV_1_ is strongly inversely correlated with haptoglobin adipocytokine;

(6) Only in the COPD group, haptoglobin/fat mass ratio, and adiponectin are significantly negatively correlated to irisin; likely, adiponectin is significantly negatively correlated to the SF-36 Physical Functioning section questionnaire. 

Our study brings additional information on the potential role of the networking among the adipocytokine leptin, adiponectin, haptoglobin, the myokine irisin, and the body composition in the daily quality of life and in the respiratory capacity of COPD patients. Systemic inflammation may also initiate or worsen comorbid diseases that can lead to impaired functional capacity, worsening dyspnea, reduced health-related quality of life, and increased mortality.

Recently, Khudiakova and co-authors in a very new study [2] investigated the associations among several adipocytokines (leptin and total adiponectin included) and chronic bronchitis in 115 young patients affected by chronic bronchitis versus 115 matched healthy controls, in presence or in absence of abdominal obesity. The authors found significantly increased leptin and total adiponectin levels in patients affected by chronic bronchitis with abdominal obesity in comparison with those without abdominal obesity. Khudiakova and co-authors concluded that their results allow to better understand the pathogenesis of the development of inflammatory diseases of the bronchi against the background of abdominal obesity. In this context, despite that our study was a pilot study performed on a total of 51 subjects between COPD and healthy controls, and that it was performed on a cohort of the population with a mean age of 67.6 years, it perfectly matches with the result of the aforementioned study, regarding leptin and the total adiponectin plasmatic levels, that we found significantly increased in COPD versus healthy controls. In addition, not only circulating leptin levels are found increased in COPD in comparison to controls, confirming what is found also in other previous studies [2,24], but also the circulating level of leptin also remains significantly strongly increased in COPD patients versus healthy controls even after normalization for the fat mass parameter (leptin/fat mass ratio).

This important result supports the hypothesis that the pleiotropic hormone leptin in COPD disease is a pro-inflammatory cytokine per se regardless of the amount of the adipose tissue, first organ of leptin production: this leads to the concept that leptin may be considered for clinical practice as a circulating new biomarker for early diagnosis and better management of COPD disease. This result is assessed only for leptin adipocytokine but not for the other tested adipocytokines. In addition, the novel adipocytokine haptoglobin has been recently assessed as a new inflammatory marker in COPD [30].

In a recent study, Higman and co-authors [52] do not find altered levels of haptoglobin in COPD patients but report a significantly positive correlation between macrophage CD163 and haptoglobin expression, supporting the role of the CD163-haptglobin in the regulation of iron bioavailability in COPD. Our results, according to the above mentioned study, reported that the serum haptoglobin level is simultaneously and significantly increased in COPD rather than in controls. In addition, its expression significantly and negatively correlated with FEV_1_ both in COPD and in controls, strongly supporting also, for haptoglobin, a role of pro-inflammatory cytokine in the immune systems of COPD patients.

On the other hand, skeletal muscle dysfunction in COPD is also related to systemic inflammation, advanced age of the patients and oxidative stress, playing an important role in COPD progression and severity [31]. In a recent study conducted in 463 COPD patients belonging to all the four categories of GOLD stages [53], it has been found that fat-free mass (FFM), hand grip strength test, BMI, and respiratory and skeletal muscle functions were significantly decreased in patients with more severe COPD, leading to the concept that the nutritional status should be considered firstly and continuously in the management of COPD progression. In this regard, smoking is the most important risk factor for the development of COPD, where exposure to cigarette smoke per se can also induce skeletal muscle dysfunction and muscle mass reduction, that could be most likely reversible by smoking cessation [33]. Skeletal muscle, as adipose tissue, is an endocrine organ secreting several hormones including the pro-myogenic factor irisin, which plays a key role in the glucose and lipid metabolism homeostasis.

Irisin, a peroxisome proliferator-activated receptor-γ coactivator 1α (PGC-1α)-dependent, is a myokine mainly produced in specific conditions like hunger, cold stimulation, and active exercise [54,55]. It acts in a paracrine way in several organs, including adipose tissue, where it induces the browning of adipose tissue from white adipocytes. Irisin was also evaluated in our pilot study and, despite several studies, indicated that its expression decreased with COPD severity. In this pilot study, we did not find any differences for irisin circulating levels between COPD patients and healthy controls. Nevertheless, we find a significant reduction in smokers in comparison to non-smoker subjects, regardless of belonging to COPD or control groups: this finding is in line with the evidence reporting skeletal muscle dysfunction and decrease in irisin level in the presence of cigarette smoking exposure [31,34]. To strengthen this result, we also find, only in COPD patients, a significant inverse correlation between both haptoglobin and total adiponectin with irisin, also normalized for the muscle mass, leading to the concept that in this cohort of patients, irisin may be counteracted by the pro-inflammatory adipocytokines, adiponectin and haptoglobin, that sustain the inflammatory state in COPD patients.

At the same time, a previous study assessed that higher plasma adiponectin levels were independently associated with emphysema, decreased BMI, female sex, older age, and lower post-bronchodilator change in FEV_1_ [56]. The combined results of symptoms of COPD, including fatigue, dizziness, and dyspnea, as well as the presence of an inflammatory state, would have contributed to limitations in physical activity. In our work, we found an inverse correlation between lower Physical Functioning score and serum total adiponectin levels only in COPD patients: this could reflect the above mentioned concept also supporting a relation between disease severity and adiponectin levels.

The inflammatory status in COPD and QoL may be affected by many conditions, such as disease activity and presence of comorbidities [57]. In this context, PhA is recognized as an indicator of cellular health and may be a useful marker of physical function in geriatric populations as well as in patients with chronic inflammation [41,58,59,60]. In previous investigations, PhA has been related with muscle strength in different conditions, poor prognosis in chronic diseases, and impaired quality of life [61,62]. Regardless of sex, age, and skeletal muscle, PhA predicts body strength, agility, and dynamic balance in healthy older adults and is positively associated with 6MWT and functional fitness. In a previous study in breast cancer patients [58], higher PhA values were found positively correlated with higher intracellular water (ICW) values, suggesting that PhA could play a role in maintaining cell membrane quality and integrity. To this regard, we found a significant positive correlation between 6MWT and each of the bioimpedance good health parameters, included PhA and ICW, supporting the previous cited study.

In addition, accordingly with Maddocks et al. [63], we found that PhA is a strong marker of exercise functional capacity, in agreement with a recent study in which the authors found that PhA was correlated to SF-36 score and 6MWT [64]. As expected, our results showed that physical functioning domain was lower in COPD patients compared to subjects without COPD. These results suggest that health status is associated to the presence of respiratory symptoms. In fact, there is an association between breathlessness and muscle weakness independent of obstructive lung function impairment, and this relationship was stronger among physically inactive subjects. It is important to note that neither the other domain of health status nor depression and anxiety were significantly different in our group. 

Similar relationships between disease severity and physical, but not mental, health status assessed by SF-36 have been reported from another population-based study [65]. It may be speculated that other factors such as older age and presence of comorbidities have an impact on overall quality of life in our groups.

Regarding I-FABP circulating level, our results confirm what a previous study reported [36], that there is no difference between COPD patients and controls, subject for I-FABP circulating plasmatic level. 

Taken together, all our evidence supports the presence of a bidirectional interconnection between adipocytokines, pulmonary function, and body composition whereby mediators produced by adipose tissue together with other inflammatory markers can be involved in the lung inflammation and contribute to the prognosis, severity, and the quality of life of COPD patients [66,67]. 

This experimental study has some limitations: 

(1) As a pilot study, it has a small sample size: this factor did not allow us to perform a normalization for the several available variables in all the comparisons; moreover, at the same time, despite this small number of subjects, the smoking statuses of the COPD group and the control group, that is significantly different (Table 1), have an evident effect: in fact, we found a significant effect of smoking on irisin/muscle mass ratio (Figure 2E); 

(2) We did not test the handgrip function nor SARC-F questionnaire for sarcopenia [68,69]; 

(3) Lastly, we tested only the I-FABP as a marker of intestinal dysbiosis/integrity enterocyte: other specific markers such as short-chain fatty acid (SCFA), as acetate, propionate, and butyrate, recently investigated in wasting diseases for lung health, were not tested. All these aforementioned markers will be under investigation in a future study in a larger cohort of COPD patients.

## 5. Conclusions

At present, evidence in literature reports that in COPD patients’ obesity is associated with risk factors for systemic chronic disease comorbidities whereas low muscle mass is associated with increased disease severity and lower quality of life [70]. Nevertheless, the prognostic utility of body composition in COPD together with the influence of both adipocytokines and myokines requires further study. 

With this pilot study, we conclude that our results may represent a new insight for better managing the treatment and to limit progression of COPD, considering new circulating biomarkers as adipocytokines together with irisin myokine and exercise tools, to reduce adiposity and systemic inflammation in the view of primary prevention.

Primary prevention by a healthy lifestyle, including not only a healthy diet but also a physical activity habit related to one’s own capacity, is the first choice, also for patients with pulmonary disease. At the present time, the real contribution of adipocytokines and myokines to skeletal muscle dysfunction in COPD have not been fully understood.

Primary prevention by lifestyle factors, including eubiosis condition, helps to maintain a general wellness of the COPD patients, by strengthening the function of immune systems and by counteracting the effect of the sedentary life and the probable consequential obesity with its low-grade state of systemic inflammation.

In this scenario, our results may lead the researchers to better investigate the role of the adipocytokines produced by adipose tissue and on the myokines produced by skeletal muscles, to better identify an effective approach for the primary prevention and for a tailored therapy, including lifestyle habits, for COPD patients.

## Figures and Tables

**Figure 1 biomolecules-13-00048-f001:**
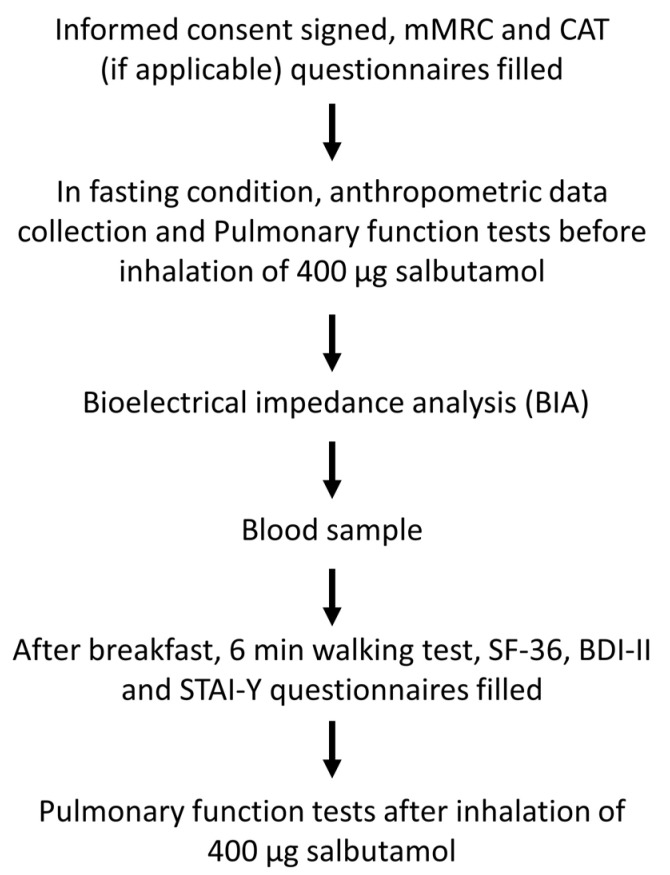
Flowchart of the sequence of data collection from the subjects who participated in the study.

**Figure 2 biomolecules-13-00048-f002:**
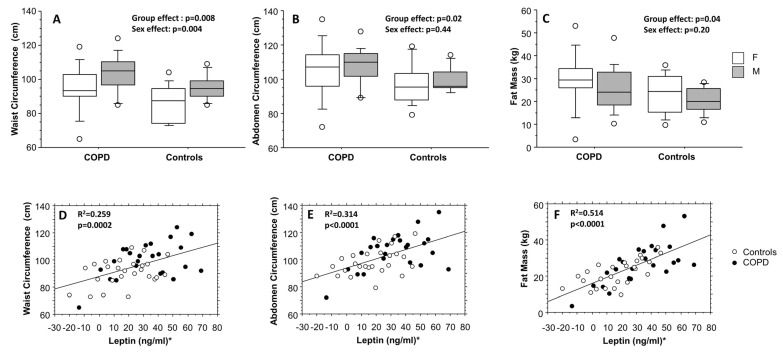
Waist circumference (**A**), abdomen circumference (**B**), and fat mass (**C**), separately for COPD and control groups and sex. Results from two-way analysis of variance. For each box plot, from the bottom to the top, horizontal lines display the 10th, 25th, 50th (median), 75th, and 90th percentiles of the values. Values below 10th percentile and above 90th percentile are plotted as circles. For both COPD and control groups, linear positive correlations between waist circumference (**D**), abdomen circumference (**E**), and fat mass (**F**) and leptin values. * Leptin values were normalized by mean and standard deviation.

**Figure 3 biomolecules-13-00048-f003:**
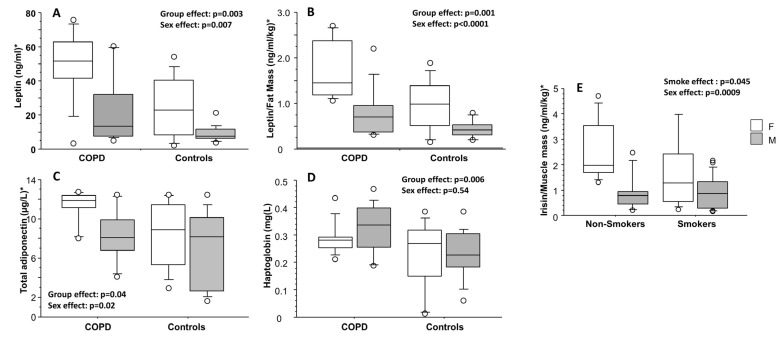
Leptin (**A**), leptin/fat mass ratio (**B**), total adiponectin (**C**), and haptoglobin (**D**) values, separately for COPD and control groups and sex. Irisin/muscle mass ratio (**E**), separately for smoking habit and sex. Results from two-way analysis of variance. For each box plot, from the bottom to the top, horizontal lines display the 10th, 25th, 50th (median), 75th, and 90th percentiles of the values. Values below 10th percentile and above 90th percentile are plotted as circles. * Leptin, leptin/fat mass ratio, total adiponectin, and irisin/muscle mass values were normalized by mean and standard deviation.

**Figure 4 biomolecules-13-00048-f004:**
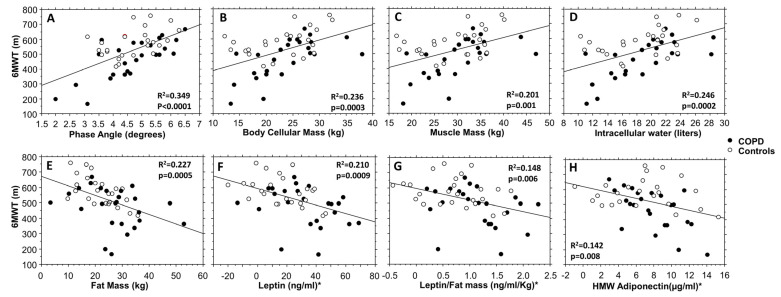
For both COPD and control groups, linear positive correlations between 6MWT and phase angle (**A**), body cellular mass (**B**), muscle mass (**C**), and intracellular water (**D**), linear inverse correlations between 6MWT and fat mass (**E**), leptin (**F**), leptin/fat mass ratio (**G**), HMW adiponectin (**H**). * Leptin, leptin/fat mass ratio, and HMW adiponectin values were normalized by mean and standard deviation.

**Figure 5 biomolecules-13-00048-f005:**
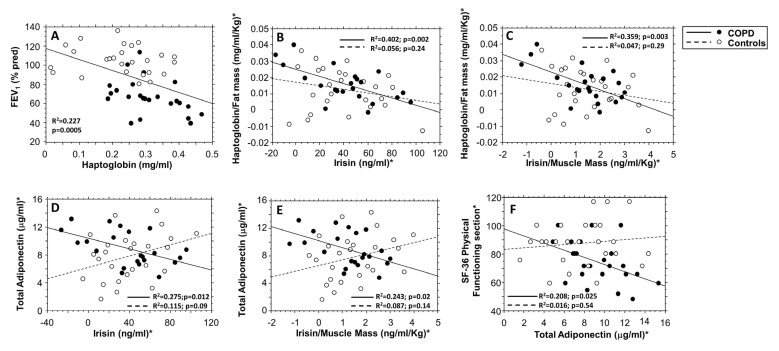
For both COPD and control groups, inverse correlations between FEV_1_ (as percent of predicted) and haptoglobin (**A**). Separately for COPD and control groups, inverse correlations between haptoglobin/fat mass and irisin (**B**), haptoglobin/fat mass and irisin/muscle mass ratio (**C**), total adiponectin and irisin (**D**), total adiponectin and irisin/muscle mass (**E**), Physical Functioning section of SF-36 questionnaire and total adiponectin (**F**). * All the variables in panels (**B**–**F**) were normalized by mean and standard deviation.

**Table 1 biomolecules-13-00048-t001:** Anthropometric and pulmonary function data, bioelectrical impedance analysis parameter evaluations, and questionnaire data of enrolled subjects.

	GROUPS		*p* Value
	COPD (No. = 25)	Controls (No. = 26)	
**Anthropometric**			
Sex (No., F/M)	10/15	12/14	0.66 #
Age (years, mean ± SD)	68.7 ± 8.6	66.5 ± 8.2	0.34 §
Weight (kg, mean ± SD)	78.6 ± 17.0	72.2 ± 12.1	0.13 §
BMI (kg/m^2^, mean ± SD)	29.8 ± 5.6	27.5 ± 3.9	0.09 §
Waist circumference (cm, mean ± SD)	100.0 ± 12.8	91.0 ± 9.8	**0.007** §
Abdomen circumference (cm, mean ± SD)	106.6 ± 13.3	98.5 ± 9.7	**0.017** §
Hip circumference (cm, mean ± SD)	107.4 ± 15.3	102.7 ± 8.4	0.18 §
Wrist circumference (cm, mean ± SD)	17.1 ± 1.4	16.5 ± 1.5	0.25 §
Smokers (No., current/former/never)	4/17/4	3/9/14	**0.02** #
**Functional**			
FVC (% predicted, mean ± SD)	88.2 ± 16.2	103.5 ± 10.0	**0.0002** §
FEV_1_ (% predicted, mean ± SD)	66.9 ± 18.0	105.4 ± 14.6	**<0.0001** §
FEV_1_/FVC (%, mean ± SD)	58.6 ± 10.4	78.9 ± 8.3	**<0.0001** §
FEF_25–75%_ (% predicted, mean ± SD)	41.1 ± 35.3	122.9 ± 54.8	**<0.0001** §
6 min walking test (m, mean ± SD)	470 ± 132	577 ± 93	**0.002** §
**Comorbidities**			
Diabetes (No., %)	3 (12%)	2 (8%)	0.61 #
Systemic hypertension (No., %)	22 (88%)	13 (50%)	**0.004** #
Dyslipidemia (No., %)	8 (32%)	10 (38%)	0.63 #
**Bioelectrical impedance analysis**			
Phase angle (degrees, mean ± SD)	4.7 ± 1.1	4.7 ± 0.8	0.94 §
Total body water (liters, mean ± SD)	40.8 ± 7.3	39.5 ± 7.4	0.54 §
Extracellular water (liters, median and IQR range)	20.4 (18.6–24.2)	20.8 (18.2–23.1)	0.72 ‡
Intracellular water (liters, mean ± SD)	18.9 ± 4.8	18.7 ± 4.6	0.86 §
Fat-free mass (kg, mean ± SD)	51.6 ± 9.5	50.5 ± 9.5	0.68 §
Body cellular mass (kg, mean ± SD)	23.6 ± 6.1	23.4 ± 6.0	0.90 §
Body cellular mass index (BCM/height^2^, median and IQR range)	8.9 ± 1.8	8.8 ± 1.6	0.88 §
Muscle mass (kg, mean ± SD)	30.1 ± 7.1	29.7 ± 7.2	0.84 §
Fat mass (kg, mean ± SD)	27.0 ± 10.9	21.8 ± 7.3	**0.046** §
**Questionnaires**			
SF-36 Physical Functioning section	75.0 (68.8–86.3)	90.0 (80.0–95.0)	**0.004** ‡
mMRC (median and IQR range)	1.0 (1.0–2.0)	0.0 (0.0–1.0)	**<0.0001** ‡
CAT (mean ± SD)	11.4 ± 3.8	N/A	N/A
BDI–II (median and IQR range)	9.0 (4.8–11.5)	8.5 (6.0–12.0)	0.96 ‡
STAI-Y (mean ± SD)	43.0 ± 9.3	43.0 ± 4.9	0.97 §

Anthropometric and pulmonary function of enrolled subjects; # χ^2^ test; § one-way ANOVA (values represent the mean ± standard deviation); ‡ Mann–Whitney U-test (values represent medians and interquartile range). SF-36, Short Form Health Survey 36; mMRC, Modified British Medical Research Council questionnaire; CAT, COPD Assessment Test; BDI–II, Beck’s Depression Inventory–II; STAI-Y, State-Trait Anxiety Inventory-Y; N/A, Not applicable.

**Table 2 biomolecules-13-00048-t002:** Adipocytokines, irisin, and I-FABP circulating levels. Two-way analysis of variance for the effects of group and sex.

	GROUPS							
	COPD (No. = 25)			Controls (No. = 26)				
	Whole Subgroup	Females (No. = 10)	Males (No. = 15)	Whole Subgroup	Females (No.= 12)	Males (No. = 14)	*p* Value for Group Effect	*p* Value for Sex Effect
Leptin (ng/mL, median and IQR range)	34.0 (10.0–58.8)	51.5 (41.8–63.0)	13.4 (7.6–32.0)	10.0 (6.2–21.0)	23.1 (8.5–40.3)	7.5 (6.2–11.7)	**0.003**	**0.007**
Leptin/fat mass (ng/mL/kg, median, and IQR range)	1.07 (0.56–1.64)	1.46 (1.18–2.37)	0.78 (0.37–0.95)	0.52 (0.34–0.84)	0.98 (0.52–1.38)	0.42 (0.31–0.53)	**0.001**	**<0.0001**
Total adiponectin (µg/mL, median, and IQR range)	8.8 (8.0–12.1)	11.9 (11.2–12.4)	8.1 (6.8–10.0)	8.4 (4.2–11.1)	8.9 (5.3–11.4)	8.1 (2.610.2)	**0.04**	**0.02**
Total adiponectin/fat mass (µg/mL/kg, median, and IQR range)	0.31 (0.23–0.50)	0.37 (0.28–0.49)	0.24 (0.23–0.50)	0.40 (0.17–0.53)	0.37 (0.19–0.45)	0.42 (0.11–0.61)	0.73	0.68
Haptoglobin (mg/mL, mean ± SD)	0.31 ± 0.08	0.28 ± 0.06	0.33 ± 0.09	0.23 ± 0.11	0.23 ± 0.13	0.22 ± 0.09	**0.006**	0.54
Haptoglobin/fat mass (mg/mL/kg, median, and IQR range)	0.011 (0.009–0.014)	0.010 (0.007–0.012)	0.011 (0.009–0.017)	0.011 (0.008–0.015)	0.010 (0.008–0.012)	0.011 (0.008–0.015)	0.77	0.09
HMW adiponectin (µg/mL, median, and IQR range)	7.38 (4.59–9.99)	9.34 (7.52–12.82)	6.11 (4.31–8.35)	6.62 (2.74–9.45)	7.09 (3.17–11.82)	6.62 (2.42–8.66)	0.13	0.07
HMW adiponectin/fat mass (µg/mL/kg, median, and IQR range)	0.22 (0.17–0.47)	0.31 (0.21–0.50)	0.21 (0.15–0.42)	0.33 (0.11–0.47)	0.32 0.16–0.40)	0.37 (0.10–0.52)	0.96	0.73
Irisin (ng/mL, median, and IQR range)	41.1 (12.7–50.3)	40.7 (13.5–61.9)	41.5 (12.8–50.6)	28.5 (10.1–49.0)	47.4 (22.6–81.9)	15.9 (9.4–33.7)	0.83	0.18
Irisin/muscle mass (ng/mL/kg, median, and IQR range)	1.22 (0.52–1.96)	1.65 (0.54–2.42)	1.07 (0.44–1.60)	0.87 (0.36–1.88)	1.89 (1.03–3.34)	0.44 (0.28–0.93)	0.67	**0.003**
I-FABP (ng/mL, median, and IQR range)	1.63 (1.15–1.98)	1.42 (1.05–1.83)	1.71 (1.19–2.00)	1.68 (1.31–2.30)	1.97 (1.42–2.35)	1.53 (1.31–2.19)	0.18	0.75

**Table 3 biomolecules-13-00048-t003:** Multiple linear regression model for 6 min walking test. R^2^ = 0.701.

	B	Lower 95%CI	Upper 95%CI	*p* Value
**Group** ** *(Ref. Controls)* **	−26.7	−98.0	44.7	0.46
**Sex (Ref. Males)**	−20.7	−72.2	30.9	0.42
**Age**	−2.7	−5.86	0.36	0.08
**FEV_1_ (% of pred.)**	0.88	−0.46	2.21	0.19
**Leptin (ng/mL) ***	−2.65	−3.91	−1.39	**<0.0001**
**Phase angle (degrees)**	57.3	27.5	87.2	**<0.0001**

Multiple linear regression model for 6 min walking test. R^2^ = 0.701; B, coefficient of multiple linear regression. * After normalization by mean and standard deviation.

## Data Availability

Data is contained within the article.

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
