# Peer review of "Relationship among Body Composition, Adipocytokines, and Irisin on Exercise Capacity and Quality of Life in COPD: A Pilot Study"

_biomolecules, 2022, doi:10.3390/biom13010048_

Round 1

Reviewer 1 Report

This pilot study investigated the roles of some biomolecules and body composition on respiratory function, functional exercise capacity and quality of life in COPD patients, and found correlations between some of these indices. It may offer some new cues for potential biomarkers regarding the characteristics of COPD. Based on data of the pilot study, further studies with larger sample sizes may be enlightened.

However, there are some points as following, which should be taken into consideration.

1.     Introduction

More background details in the measurements of body composition and the relationship between these indices and obesity, and muscle dysfunction and/or impaired muscle mass are needed.

Information about I-FABP is missing in this part as well.

2.     Design

The title is focused on “adipocytokines and irisin in the … assessment of body composition”, but the study was designed to investigate and did find more, especially in respiratory function, functional exercise capacity and quality of life, etc. A more informative title is suggested.

Was there any consideration in the interferences to the results from different stages of COPD severity, duration from the onset of COPD, different therapies administered, and/or age? As subgroups? Older patients may be more susceptible to worse nutritional status, skeletal muscle dysfunction (Page 13, Line 425) and sarcopenia, and higher plasma adiponectin levels were associated with older age (Page 13, Line 453). The fat free mass, hand grip strength test, and BMI were decreased in patients with COPD in advanced stages (Page 12, Line 428).

Were the blood parameters of lipid profile and glucose taken into consideration? Were the patients with diabetes, hyperlipidemia, and metabolic syndrome excluded?

3.     Methods

How was the quality of collecting data of subjective scales or questionnaires controlled? Were patients with dementia and/or other cognitive deficiencies excluded? Were there guidance or participation from professional psychiatric physician, especially in evaluating depression and anxiety?

A flow chart of the sequences and time points at which the indices or parameters were measured is suggested, and every patient should be tested according to it. Irisin can be produced after taking physical activities (Page 13, line 437-9). Were the blood samples for assessing irisin collected after the walking tests in some of the patients? Not in the others?

How was the control group matched?

Were all the correlations between each two possibly meaningful indices measured analyzed, and only those with statistically significant results reported for publishing?

4.     Results

Table 1 and Table 2 are suggested to be merged into one table, followed by Figure 1. Description in Fig 1 and Fig 2 can be more consistently (“class effect” in Fig 1 while “group effect” in Fig 2).

In Table 4, there were possible correlations between some variables (e.g. leptin and category or sex, according to the former results in this study). Were there interactions between each two and needed to be excluded?

Are there more potentially meaningful negative findings considered for representation and interpretation?

5.     Discussion and conclusion

Is there an explanation for the difference between the results in adiponectin and HMW adiponectin (Table 3)? And the discussion of the variable “Category” (COPD or Controls) being not significant in the regression model for 6MWT in Table 4 (caused by the limitation of small sample size, insensitivity of 6MWT test in these patients, etc?)? The sex effects in Table 3 and Figure 2?

There were no dynamic observations of leptin and time-related indicator measurement in this study, and caution is needed to draw inferences that “leptin…for early diagnosis…of COPD” (Page 12, Line 412). The result of this study can not offer information of the causal relationship between these biomolecules, body components, obesity, muscle loss, and COPD. The use of these biomolecules as biomarkers for the prevention, diagnosis and treatment of COPD has yet to be studied.

Author Response

Reviewer 1

Comments and Suggestions for Authors

This pilot study investigated the roles of some biomolecules and body composition on respiratory function, functional exercise capacity and quality of life in COPD patients, and found correlations between some of these indices. It may offer some new cues for potential biomarkers regarding the characteristics of COPD. Based on data of the pilot study, further studies with larger sample sizes may be enlightened. However, here are some points as following, which should be taken into consideration.

We thank the Reviewer for her/his careful review of our work, and we have done our best to answer the majors’ concerns.

  1. Introduction

More background details in the measurements of body composition and the relationship between these indices and obesity, and muscle dysfunction and/or impaired muscle mass are needed. Information about I-FABP is missing in this part as well.

We thank the Reviewer for her/his right comments. The new concepts and the related new references n 32 and n 37 (Machado, F.V.C.; Schneider, L.P.; Fonseca, J.; Belo, L.F.; Bonomo, C.; Morita, A.A.; Furlanetto, K.C.; Felcar, J.M.; Rodrigues, A.; Franssen, F.M.E.; et al. Clinical Impact of Body Composition Phenotypes in Patients with COPD: A Retrospective Analysis. Eur. J. Clin. Nutr. 2019, 73, 1512–1519, doi:10.1038/s41430-019-0390-4 and McKenna, Z.; Houck, J.; Ducharme, J.; Li, Z.; Berkemeier, Q.; Fennel, Z.; Wells, A.; Mermier, C.; Deyhle, M.; Laitano, O.; et al. The Effect of Prolonged Interval and Continuous Exercise in the Heat on Circulatory Markers of Intestinal Barrier Integrity. Eur. J. Appl. Physiol. 2022, 122, 2651–2659, doi:10.1007/s00421-022-05049-4..) about the body composition and the relationship among obesity, muscle dysfunction and impaired muscle mass (102-111, lines) and I-FABP introduction (115-123, lines) are now in the new revised manuscript.

  1. Design

The title is focused on “adipocytokines and irisin in the … assessment of body composition”, but the study was designed to investigate and did find more, especially in respiratory function, functional exercise capacity and quality of life, etc. A more informative title is suggested.

We thank the Reviewer. Now, we proposed this title: Relationship among body composition, adipocytokines and irisin on exercise capacity and Quality of Life in COPD: a pilot study.

Was there any consideration in the interferences to the results from different stages of COPD severity, duration from the onset of COPD, different therapies administered, and/or age? As subgroups? Older patients may be more susceptible to worse nutritional status, skeletal muscle dysfunction (Page 13, Line 425) and sarcopenia, and higher plasma adiponectin levels were associated with older age (Page 13, Line 453). The fat free mass, hand grip strength test, and BMI were decreased in patients with COPD in advanced stages (Page 12, Line 428).

The precise role of adiponectin in the pathophysiology of COPD is still unclear because adiponectin is reported to exert both proinflammatory and anti-inflammatory effects in patients affected by COPD. One possible explanation for the proinflammatory hypothesis could be that increased inflammation in COPD leads to an increase in circulating level of total adiponectin as a pro-inflammatory molecule.  Furthermore, total adiponectin exerts its proinflammatory effects on airway epithelium or fat cells and that the resulting altered metabolism contributes to functional loss of lung tissue (i.e., emphysema), implicating adiponectin as a phenotypic biomarker of COPD. (Carolan, B. J., Kim, Y. I., Williams, A. A., Kechris, K., Lutz, S., Reisdorph, N., & Bowler, R. P. (2013). The association of adiponectin with computed tomography phenotypes in chronic obstructive pulmonary disease. American journal of respiratory and critical care medicine, 188(5), 561-566.). The aim of our study was also to better elucidate the role of total adiponectin in COPD. Thanks to the comments of the Reviewer, we perform the new correlation analyses for total adiponectin and the different stages of COPD severity, the duration from the onset of COPD, and the age of the subjects enrolled. Due to the small sample size, we could not perform any statistical analysis for the different administered therapies. As concern of COPD severity, no effect of COPD severity was found on Total adiponectin (Figure 1 below). Also, this result may be due to the limited sample size in our pilot study. and to the lack of stage 4 patients. Hence, in contrast to our findings Chan et al. founded a higher plasma adiponectin level in patients with more severe COPD suggesting that adiponectin might be inappropriately secreted in this disease. (Chan, K. H., Yeung, S. C., Yao, T. J., Ip, M. S. M., Cheung, A. H. K., Chan-Yeung, M. M. W., ...& COPD Study Group of the Hong Kong Thoracic Society. (2010). Elevated plasma adiponectin levels in patients with chronic obstructive pulmonary disease. The International journal of tuberculosis and lung disease, 14(9), 1193-1200.). Regarding a possible effect of disease duration, we did not find any correlation between normalized Total adiponectin and disease duration (in years) (Figure 2). Conversely, as concern of age, we found a significant effect on normalized Total adiponectin. In a linear correlation model, Total adiponectin and age showed a clear negative correlation in Control group only, without any significant correlation among COPD (Figure 3). A possible explanation for the lack of such correlation among COPD patients could be the “confounding” effect produced by the chronic inflammation of these patients. Please, find below the figures related to the new described correlations.

Figure 1 – Box plot of Total adiponectin levels separately for COPD severity (GOLD Stages 1-3). p=0.854, Kruskal-Wallis test. For each box plot, from the bottom to the top, horizontal lines display the 10th, 25th, 50th (median), 75th, and 90th percentiles of the values. Values below 10th percentile and above 90th percentile are plotted as circles

Figure 2 - Linear correlation model between Total adiponectin and Disease duration in COPD Patients

Figure 3 - Linear correlation model between Total adiponectin and age, separately for COPD and Controls

Were the blood parameters of lipid profile and glucose taken into consideration? Were the patients with diabetes, hyperlipidemia, and metabolic syndrome excluded?

We did not take into any consideration individual lipid profile or glucose plasmatic levels. Regarding the comorbidities mentioned by the Reviewer, in our experimental design we did not exclude subjects with comorbidities as treated diabetes, systemic hypertension or dyslipidemia. The frequency distribution of comorbidities reported by Subjects and their significant differences between the study groups are now reported in Table 1 in the new manuscript version. Neither dyslipidemia nor diabetes had a significantly different frequency distribution between groups. Only systemic hypertension showed higher prevalence among COPD.

  1. Methods

How was the quality of collecting data of subjective scales or questionnaires controlled? Were patients with dementia and/or other cognitive deficiencies excluded? Were there guidance or participation from professional psychiatric physician, especially in evaluating depression and anxiety?

At an initial visit, all patients received information about the rationale and objectives of the research; we verified eligibility criteria by reviewing their inclusion and exclusion criteria and recorded general socio-demographic characteristics. Patients with dementia and with other cognitive deficiencies were excluded from this study. This is reported in the new version of the manuscript (150, 151 lines). Patients completed auto-administered patient questionnaires. All data were collected as part of the clinic’s routine diagnostic procedures approved by the ethic committee. An experienced clinical psychologist explained the procedures inherent in the auto-administration of quality of life, anxiety and depression questionnaires. The details of the exclusion criteria are now in the new version of manuscript.

A flow chart of the sequences and time points at which the indices or parameters were measured is suggested, and every patient should be tested according to it. Irisin can be produced after taking physical activities (Page 13, line 437-9). Were the blood samples for assessing irisin collected after the walking tests in some of the patients? Not in the others?

We thank the Reviewer for her/his comments and the right observations. The flow chart is presented in the new version of manuscript at the end of “Methods” section as Figure 1. Based also on flow chart, irisin evaluation was made in serum from fasting subjects, COPD and Healthy Controls all in the same conditions (266 line).

How was the control group matched?

We thank the Reviewer. Healthy control subjects were matched by age and BMI, keeping the sex distribution not significantly different between COPD and control subjects. The description is in the new version of the manuscript, (145-147, lines).

Were all the correlations between each two possibly meaningful indices measured analyzed, and only those with statistically significant results reported for publishing?

We thank the Reviewer for her/his comments and the right observations: we performed several correlation analyses, please find below the analysis performed that are not significant and that could be also included in the definitive version of the manuscript.

As concerns the not significant correlations we found in our study, we evaluated (not presented in the manuscript) the following:

Correlation between 6 minutes walking test and Haptoglobin levels, in COPD and Control subjects (not significant)

Correlation between Leptin levels and Phase angle, separately for COPD and Controls (both not significant)

Correlation between Irisin levels and Phase angle, separately for COPD and Controls (both not significant)

  1. Results

Table 1 and Table 2 are suggested to be merged into one table, followed by Figure 1. Description in Fig 1 and Fig 2 can be more consistently (“class effect” in Fig 1 while “group effect” in Fig 2).

We thank the Reviewer, now we identify both in Figures 1 and Figures 2 (and throughout the text) as “group effect” and not as “class  effect”. Table 1 and 2 were merged, as requested, in new Table 1

In Table 4, there were possible correlations between some variables (e.g. leptin and category or sex, according to the former results in this study). Were there interactions between each two and needed to be excluded?

As shown in Table 3 and Figure 2 (panel A), leptin was significantly different between groups (COPD and Controls) and presenting a significant sex effect. In the relevant two-way ANOVA model, the interaction between Group and Sex was not significant (please find below the figure). Nevertheless, in the multiple regression model presented in Table 4, the interaction between leptin and sex and between leptin and Group were not significant

p value for Group effect = 0.0026

p value for Sex effect = 0.0067

p value for interaction = 0.628

Are there more potentially meaningful negative findings considered for representation and interpretation?

As we wrote before, we performed several correlation analyses that are not significant, that could be also included in the definitive version of the manuscript.

  1. Discussion and conclusion

Is there an explanation for the difference between the results in adiponectin and HMW adiponectin (Table 3)? And the discussion of the variable “Category” (COPD or Controls) being not significant in the regression model for 6MWT in Table 4 (caused by the limitation of small sample size, insensitivity of 6MWT test in these patients, etc?)? The sex effects in Table 3 and Figure 2?

We thank the Reviewer for her/his right observation. Actually, as already we discussed for total adiponectin, which role is still controversial in COPD, the same concept we can discuss for the High Molecular Weight (HMW) adiponectin. Further studies with a larger sample size of subjects enrolled are needed to explain this difference. Indeed, the result presented in Table 4 (Table 3 in the new version) is somewhat surprising: in fact, when in the model are introduced the two variables Leptin and Phase angle (significantly correlated to 6 minutes walking test – negatively the first, positively the other one [Figure 4]) it appears that both “drive” the effect on the outcome variable (6MWT), irrespectively of the Group (COPD and Control). This could suggest that the good health nutritional parameters Phase angle (PhA) and leptin, as pro-inflammatory factor per se independently from the fat mass, affect the physical performance, both in COPD and in healthy control subjects. Regarding the Table 3 (now Table 2 in the new version of the manuscript) we introduced the column for the whole subgroup (females and males together) to let the table clearer to the reader regarding the sex effects already shown both in the Table 3 (now Table 2) and in Figure 2 (now Figure 3 in the new version of the manuscript). We hope now the description is clearer.

There were no dynamic observations of leptin and time-related indicator measurement in this study, and caution is needed to draw inferences that “leptin…for early diagnosis…of COPD” (Page 12, Line 412). The result of this study can not offer information of the causal relationship between these biomolecules, body components, obesity, muscle loss, and COPD. The use of these biomolecules as biomarkers for the prevention, diagnosis and treatment of COPD has yet to be studied.

We thank the Reviewer for her/his comments. This comment is the main final message of  our pilot study and it is perfectly in line with the state of art of leptin in COPD. With our pilot study we just hypothesize that leptin adipocytokine could be considered as a new circulating biomarker in COPD as it has been largely studied as a pro-inflammatory cytokine in the last 20 years in lung diseases and in COPD, specifically. Based on this, in the conclusion of the manuscript we indicated that “With this pilot study, we conclude that our results may represent a new insight to better manage the treatment and to limit progression of COPD considering new circulant biomarkers as adipocytokines together with irisin myokine and exercise tool, to reduce adiposity and systemic inflammation in the view of primary prevention”. Based on this context, we aim that the clinical researcher could hypothesize leptin as a new circulating biomarker for COPD diagnosis and management and could plan the future research and the next experimental clinical studies in this direction. 

Reviewer 2 Report

This study endeavors to investigate the relationships between circulating levels of irisin, haptoglobin, and the adipokines, leptin and adiponectin, and a variety of clinical and quality of life parameters data collected in patients with COPD and healthy controls. The goal was to address the hypothesis that modulation of production of the factors may contribute to COPD pathogenesis. The authors found some significant correlations of serum levels of the factors with COPD in general and with physical function in COPD as determined by questionnaires.

The study design is interesting, but I have several major comments that would strengthen the manuscript.

1)    The writing of the manuscript would be improved by proofreading for appropriate English grammar, punctuation and sentence structure. The writing is characterized by many long, run-on sentences that are difficult for the reader. Also, the introduction and Discussion sections should be broken into paragraphs. As such, it is difficult for the reader to discern the major findings and significance of the study.

2)    The study states that healthy subjects are “matched” with the COPD, but the criteria for matching are not described. Are the patients matched based on age? BMI? Please clarify the criteria for matching.

3)    The smoking statuses of the COPD group and the control group are significantly different, which may confound the data substantially in such a small study. Please address this limitation in the Discussion and state whether the data were normalized for smoking status in each of the analyses.

4)    Clarity is needed in representation of the data in figures and tables.

a.     In figures showing bar graphs (Figures 1A,B,C; Figures 2D,E), it would be helpful to show the data points superimposed on the bars so that the reader can see the number of data points in each analysis and the spread of the data.

b.     For figures that show data separated by male and females, it is also important to show additional graphs or add data to the tables for the combined male/female data, which is depicted in the “class effect” (Figure 1A,B,C) or “group effect” (Table 3; Figure 2 A-E) values listed on the figures and tables.

c.     In Figure 4, the legend should also list which best-fit line (dashed or solid) is associated with each experimental group (COPD or controls)

Author Response

Reviewer 2

Comments and Suggestions for Authors

This study endeavors to investigate the relationships between circulating levels of irisin, haptoglobin, and the adipokines, leptin and adiponectin, and a variety of clinical and quality of life parameters data collected in patients with COPD and healthy controls. The goal was to address the hypothesis that modulation of production of the factors may contribute to COPD pathogenesis. The authors found some significant correlations of serum levels of the factors with COPD in general and with physical function in COPD as determined by questionnaires.

The study design is interesting, but I have several major comments that would strengthen the manuscript.

We thank the Reviewer for her/his careful review and for her/his appreciation of our work. We have done our best to fully answer to her/his comments.

  • The writing of the manuscript would be improved by proofreading for appropriate English grammar, punctuation and sentence structure. The writing is characterized by many long, run-on sentences that are difficult for the reader. Also, the introduction and Discussion sections should be broken into paragraphs. As such, it is difficult for the reader to discern the major findings and significance of the study.

We fully agree with the Reviewer. In the new version of the manuscript both the introduction and the discussion were broken in subparagraphs. Furthermore, we tried to shorten the longest sentences to allow the study clearer to the reader. Lastly, we committed the correction of the whole manuscript for appropriate English grammar, punctuation and sentence structure to an English mother-tongue teacher (Mrs Lisa Prest a native English teacher). Her name is written in the new version of the manuscript, in the acknowledgment section.

  • The study states that healthy subjects are “matched” with the COPD, but the criteria for matching are not described. Are the patients matched based on age? BMI? Please clarify the criteria for matching.

We thank the Reviewer for her/his comments and the right observations. The description is in the new version of the manuscript, (145-147, lines).

  • The smoking statuses of the COPD group and the control group are significantly different, which may confound the data substantially in such a small study. Please address this limitation in the Discussion and state whether the data were normalized for smoking status in each of the analyses.

Thanks for your comments. We included in the first limitation of our pilot study, at the end of Discussion, this concept and we explain that, despite the small size, the effect of smoking status is evident in our sample. In fact, we found a significant effect of smoking on irisin/muscle mass ratio (Figure 2, panel E).Following,  we would like to present to the Reviewer two comparisons relevant to the possible effect of smoking habit on Total adiponectin and on haptoglobin. Both the comparisons resulted not significant. Please, find these graphs below.

At the same time, unfortunately, the sample size did not allow a normalization for the several available variables in all the comparisons: this is now acknowledged among study limitations. We hope that now the new version of the manuscript is clearer

4)    Clarity is needed in representation of the data in figures and tables.

  1. In figures showing bar graphs (Figures 1A,B,C; Figures 2D,E), it would be helpful to show the data points superimposed on the bars so that the reader can see the number of data points in each analysis and the spread of the data.

We thank the Reviewer for her/his observation: accordingly, we changed the indicated panels in Figures 1 and 2 (now Figures 2 and 3) from bar plots to box plots which better show the data distribution.

  1. For figures that show data separated by male and females, it is also important to show additional graphs or add data to the tables for the combined male/female data, which is depicted in the “class effect” (Figure 1A,B,C) or “group effect” (Table 3; Figure 2 A-E) values listed on the figures and tables.

Thanks for your comments Also following the observation raised by Reviewer No. 1, now COPD and Controls are called “Groups”. A new Table 3 (now Table 2) is provided showing not only the values by sex but also for the whole subgroups (COPD and Controls).

  1. In Figure 4, the legend should also list which best-fit line (dashed or solid) is associated with each experimental group (COPD or controls)

We thank the Reviewer for her/his correct observation, the legend was missing in Figure 4. It is now added (now it is Figure 5) in the upper left corner of the new version of manuscript.

Round 2

Reviewer 2 Report

This revised manuscript is much improved, and the authors have responded satisfactorily to my concerns. I have no further revisions to suggest. 

Author Response

Thanks